# Assessing Environmental Factors within the One Health Approach

**DOI:** 10.3390/medicina57030240

**Published:** 2021-03-05

**Authors:** Sarah Humboldt-Dachroeden, Alberto Mantovani

**Affiliations:** 1Department of Social Science and Business, Roskilde University, Universitetsvej 1, 4000 Roskilde, Denmark; 2Department on Food Safety, Nutrition and Veterinary Public Health, Italian National Institute of Health (ISS), 00161 Roma, Italy; alberto.mantovani@iss.it

**Keywords:** one health, environment, antimicrobial resistance, DANMAP, mycotoxins, risk analysis, food safety, farm to fork

## Abstract

*Background:* One Health is a comprehensive and multisectoral approach to assess and examine the health of animals, humans and the environment. However, while the One Health approach gains increasing momentum, its practical application meets hindrances. This paper investigates the environmental pillar of the One Health approach, using two case studies to highlight the integration of environmental considerations. The first case study pertains to the Danish monitoring and surveillance programme for antimicrobial resistance, DANMAP. The second case illustrates the occurrence of aflatoxin M1 (AFM1) in milk in dairy-producing ruminants in Italian regions. *Method:* A scientific literature search was conducted in PubMed and Web of Science to locate articles informing the two cases. Grey literature was gathered to describe the cases as well as their contexts. *Results:* 19 articles and 10 reports were reviewed and informed the two cases. The cases show how the environmental component influences the apparent impacts for human and animal health. The DANMAP highlights the two approaches One Health and farm to fork. The literature provides information on the comprehensiveness of the DANMAP, but highlights some shortcomings in terms of environmental considerations. The AFM1 case, the milk metabolite of the carcinogenic mycotoxin aflatoxin B1, shows that dairy products are heavily impacted by changes of the climate as well as by economic drivers. *Conclusions:* The two cases show that environmental conditions directly influence the onset and diffusion of hazardous factors. Climate change, treatment of soils, water and standards in slaughterhouses as well as farms can have a great impact on the health of animals, humans and the environment. Hence, it is important to include environmental considerations, for example, via engaging environmental experts and sharing data. Further case studies will help to better define the roles of environment in One Health scenarios.

## 1. Introduction

One Health is a concept that has gained popularity during the last years, especially since the Tripartite engagement of the World Health Organization (WHO), the Food and Agriculture Organization (FAO) and the World Organization for Animal Health (OiE) in 2010 [1]. The Tripartite defines One Health as:

“An approach to address a health threat at the human-animal-environment interface based on collaboration, communication, and coordination across all relevant sectors and disciplines, with the ultimate goal of achieving optimal health outcomes for both people and animals; a One Health approach is applicable at the subnational, national, regional, and global level.”[2]

Infectious zoonotic diseases are a main One Health issue, as these diseases transmit from animals to humans and vice versa. The environment, where humans interact with farm animals, pets or wild animals, plays an important role for disease transmission. The ecosystem and how it is shaped by human activities like agriculture, is an important determinant for the risk assessment of zoonoses transmitted by wildlife [3]. Climate change represents a crucial example of an environmental factor severely impacting wild and domestic animal populations, food chains and human health [4,5,6]. Changes of the climate like altering temperatures can play a considerable role in the spread of diseases. It can affect the migration and adaptation of infectious pathogens like bacteria, viruses, parasites and fungi. Through climate change, infectious pathogens may find new habitats, which can cause diseases in new and previously unaffected geographical regions [7]. Mycotoxin-producing fungi are an example of plant pathogens, whose incidence is modified by climate changes. Among mycotoxins, aflatoxins are especially poisonous and these naturally occurring toxins may contaminate feed and food and adversely affect animal and human health [8]. Further, the carry-over of pollutants from farm animals to human food is influenced by the environment as well as by the animal metabolism, and it is associated to health risks for humans consuming foods of animal origin and also for animals [9].

Antimicrobial resistance (AMR) is another topic that points out the connectedness of animals, humans and the environment. It is a global concern as it threatens the ability to treat infections in humans, animals and plants. AMR occurs when microbes such as bacteria, fungi, viruses or parasites change so that conventional treatments fail. Factors that increase the selective pressure toward resistant pathogenic strains are misuse and overuse of antimicrobial drugs in humans and livestock; inadequately tested antimicrobial pesticides for plants; inadequately enforced agricultural regulations; as well as insufficient awareness. AMR can spread between humans and animals and circulate through the environment; for example, via food products [10]. The presence of toxic metals in the environment, such as arsenic or copper, can also enhance AMR by eliciting bacterial co-resistance or cross-resistance mechanisms [11].

The increasing threat of zoonoses and AMR highlight the importance of a One Health approach able to cope with complex, multifaceted problems. While the One Health approach evolved especially since the Tripartite engagement of the WHO, FAO and OiE in 2010, similar approaches like farm to fork have been introduced, too [1]. The farm-to-fork strategy was implemented by the European Union to guarantee food safety, integrating sustainable food systems [12]. In particular, the strategy calls for a One Health perspective applied to scientific opinions and intends to support an up-to-date regulatory framework: the risks to human health are considered alongside the health of food-producing organisms and the potential impact of food chain components on the environment, such as substances used in animal feed [13].–In recognition of the need to effectively tackle complex problems, the One Health approach is now widely appreciated for interdisciplinary research and is integrated with the farm-to-fork strategy, currently considered in high-level strategic documents [12,14]. For example, the report by the European Commission, “A European One Health Action Plan against Antimicrobial Resistance”, is based on the One Health approach, mentioning the importance of considering the human–animal–environment interface [15]. The farm-to-fork approach is implemented as a strategy for the European Green Deal, a plan developed in 2020 to make Europe climate neutral by 2050. The plan also promotes One Health in the context of AMR and sustainable food production [12].

However, while the One Health approach gains increasing momentum because of its multifaceted aspects and due to the Covid-19 pandemic, its practical application meets hindrances [16]. One Health implementation calls for identifying priority areas for added value of joint activities, and for the effective knowledge elicitation of experts from different and relevant disciplines. Accordingly, One Health may call for updated models for establishing and maintaining effective and timely collaboration and communication across and within disciplines. The establishment of One Health approaches and networks can be of high value for countries that aim to establish or improve their One Health activities, for instance to support science-based regulations in the fields of health, food and environment [2,17].

In the evolving One Health field, there are gaps, open questions and challenges about meaningful integration and institutionalisation of the approach [18]. Zoonoses have been the cradle of One Health; therefore, human–animal relationships have had an ample impact as the first two pillars of the One Health approach [4]. Much thought and actions are needed to optimise the role of environment as the third pillar. Main challenges include how environmental datasets and factors can strengthen the One Health approach for issues such as AMR, as well as how to assess environmental and health issues such as toxic pollutants [17,19].

### One Health and the Environment

When reviewing One Health activities, veterinary as well as medical themes prevail and the environment is often neglected [20]. Nevertheless, the environment is all around us, it depends on and affects human and animal health in many ways. For example, healthy soils and clean water can prevent the spread of diseases, and clean environments in slaughterhouses, preservation of natural habitats of animals and biodiversity can contribute to fewer disease infections in animals and humans [21,22,23,24]. Climate change is another perspective demonstrating ecological changes affecting environmental, animal and human health. Zinsstag et al. displayed how One Health considerations can aid in solving issues resulting from climate change, such as livestock farming, food security, food safety and sanitation. Integrating public health concerns as well as animal and environmental health perspectives can contribute to enhanced and more contextual problem solving [25]. Yet, beyond the recognition of the importance of the environmental pillar, in what scenarios can it be integrated?

In the following and displayed in Table 1, we propose two partly overlapping scenarios, describing different environmental impacts that highlight the importance of the environment within the One Health approach:(I)Environmental changes modulating risk factors for health

A good example for environmental changes is provided by climate change. Events driven by climate changes may increase the availability of toxicants for food-producing organisms: erosion of soils from flooding, heavy rainfall, thawing of frozen soil and forest fires release mercury from “trapping” environments into the ecosystem [26]. Factors such as temperature and humidity affect the availability of toxic pollutants like lead, causing adverse health effects in animals and humans [27]. It can also aid the distribution of some zoonotic vectors, which in turn affect disease epidemiology. Climatic changes may affect some regions and some populations more than others. More data are needed for a thorough risk assessment, since drivers of vector populations show specific patterns according to vector species and regions [7]. The ongoing and developing scenario of the Covid-19 pandemic highlights how the health impact of an infectious disease can be modulated by a number of diverse, environment-related factors, including meteorological conditions, air pollutants, sewage and wastewater management and even by industrial chemicals, which are widespread, persistent and immunotoxic [28,29,30,31].

(II)
*Anthropogenic activities as a source of One Health risk factors through the environment*


Anthropogenic activities are main drivers that shape the environment [32,33]. Environment-modifying human activities include improper disposal of toxic waste, impacts of industrial emissions, utilisation of polluted wastewater or manure on pastures and crops used as animal feed. The presence of zoonotic agents in manure is a recognised problem, and methods for anaerobic digestion and manure storage are envisaged to reduce the potential risks [34]. Some pollutants may bioaccumulate in farm animals, and the human exposure is mediated and modified through the animal metabolism and ecology. An example of the industrial impact of exposure to pollutants is the persistent and bioaccumulating β-hexachlorocyclohexane, a by-product of the insecticide lindane. In an instance in Italy, the insecticide accumulated in industrial waste was found in animals, feed and humans [35]. Lifestyle choices and food habits were important predictors of human exposure to the insecticide, which highlights the importance of a One Health perspective [36]. Concerning pesticides, the European Food Safety Authority (EFSA) has recommended approaches beyond the characterisation of hazards and towards the risk assessment of different ecosystems through the integration of datasets coming from disciplines like ecology, biology and toxicology [37]. Intriguingly, the intensive use of herbicides such as glyphosate and glufosinate is suggested to increase the selective pressure towards antibiotic resistance in environmental bacterial communities, indicating yet another link between chemical pollution and a typical One Health issue such as AMR [38].

As highlighted above, it appears that the environmental pillar of One Health is evident, yet, case studies are needed to assess and exploit the environmental component in a One Health-based risk analysis.

This study presents two cases to portray the importance of environmental considerations in the One Health interface. The abovementioned scenarios are used as an orientation under which the cases are arranged. The two cases describe the Danish Integrated Antimicrobial Resistance Monitoring and Research Programme (DANMAP) and the environment-driven impact of the mycotoxin aflatoxin M1 on dairy farming in Italy.

## 2. Methods

Two cases were identified to exemplify One Health approaches with environmental considerations. The first case describes the DANMAP. A scientific and grey literature review was conducted to locate relevant articles and documents to describe the case. For this purpose, the database Web of Science was searched. The literature search included relevant articles in English from 1995, the year in which DANMAP was implemented, until January 2021. Keywords used for the search for the DANMAP case included the terms “DANMAP”, “AMR”, “Antimicrobial resistance”, “from farm to fork”, “One Health”. Included were English articles that mentioned DANMAP or articles containing themes pertaining to antimicrobial resistance in Denmark. Disregarded were articles that presented research on specific pathogens or scientific methods.

The grey literature search gathered DANMAP reports and additional information on the Danish and European antimicrobial resistance approach. The grey literature was located through a web search via Google. DANMAP reports were found on the website created for the programme (https://www.danmap.org/ (accessed on 27 January 2021)). The reports are mainly in English, but the report from 1997 written in Danish was also included. Additional sources were either found through references in the reports or internet searches.

For Aflatoxin M1, a literature search was performed in PubMed using the search term “Aflatoxin M1” and (“dairy” or “cheese” or “milk”). From this search, two subsets were extracted, using as search terms “climate” and “Italy”. The search included English articles published from the early 2000s, when the first aflatoxin case in Italy occurred, until January 2021.

Datasets from an EFSA opinion on aflatoxins and by the Italian Food Safety National Committee were also used [39,40]. Grey literature pertaining to aflatoxin-related issues were obtained by an internet search.

## 3. Results

The literature search revealed scientific articles and reports that aided the analysis of the two case studies. The search identified 294 articles, of which 28 articles were included into the analysis; see Figure 1. In total, 266 articles were excluded based on a screening of title and abstract and a subsequent full-text screening of the remaining articles. Additionally, ten reports were found and included in the analysis. The articles were used to explain the cases and their backgrounds. The reports were used for a more general understanding of European and international One Health perspectives, as well as in-depth analysis of DANMAP reports.

### 3.1. The Danish AMR Monitoring Programme

Denmark implemented the AMR monitoring programme DANMAP to tackle the challenges of AMR in 1995, and it was the first country to do so [41]. The programme was initiated by the Danish Ministry of Food, Agriculture and Fisheries and the Danish Ministry of Health. DANMAP is funded by the Ministry of Health and the Ministry of Environment and Food and is performed by the Public Health Institute (Statens Serum Institut), responsible for human health aspects and the National Food Institute, responsible for food and veterinary sections. In 2004, the Danish Veterinary Institute was fused with the National Food Institute, which might be the reason that the National Food Institute covers veterinary themes [42]. The DANMAP presents four objectives: the first two relate to (1) monitoring presence of antimicrobial residues in food and feed as well as (2) the occurrences of AMR in bacteria. The latter two concern association with (3) antimicrobial consumption, transmission routes and (4) potential further research areas [43].

The DANMAP has since produced scientific knowledge on AMR and it focuses on the collaboration between the human, food and veterinary sectors, but also includes other stakeholders like farmers, slaughterhouses and pharmacies [44]. Since the initiation of the programme, Denmark has successfully reduced the prevalence of AMR [10]. Most of the attention within DANMAP is provided by the public health and veterinary health sectors; nevertheless, the environment was included. For DANMAP, the environment includes the areas where humans and animals meet, shelters of animals and places that are susceptible for infection [44]. Already in the first report of 1996, food and environmental laboratories were involved in analysing food samples from animal and non-animal origin, such as fruits and vegetables [45]. The surveillance results of bacteria in 1996 found resistance of antimicrobials in the environment [45,46]. In the 1997 report, the occurrence of resistance among *Escherichia coli* from fruit and vegetables was also found for eight antibiotics. No further specifications of actions or implications were mentioned in the report. While in the 1996 and 1997 reports, fruit and vegetable sampling was mentioned, it was not mentioned in the DANMAP reports after 1997.

For most of the years from 1997, the DANMAP reports have mentioned the approach from farm to fork and it was integrated into the AMR surveillance activities [47,48]. The strategy is utilised to monitor the entire food chain and further, as they state: “from farm to fork to patient” [44].

Since 2010, the One Health approach has been incorporated in the reports and emphasised as a fundamental principle when monitoring and researching AMR [49]. Although the One Health concept is mentioned, the reports often fail to explicitly mention the environment sector. Nevertheless, DANMAP acknowledges that the environment can be the source of infection, as exemplified by showing environmental transmission routes of different bacteria in the 2019 report [49]. Additionally, the environment is acknowledged as a source of exposure to antimicrobials and to AMR-carrying bacteria for both animals and humans. Hygiene and biosecurity measures are therefore endorsed in immediate environments of farms and hospitals [44]. To accompany the DANMAP reports, the Danish government published a One Health strategy to tackle antibiotic resistance in 2017. It contains five goals of which at least three are relevant for environmental considerations within antibiotic resistance. The first goal, “A prudent use of antibiotics to reduce the incidence of resistance”, emphasises that the environment can be a reservoir for microbes and can transmit microbes to animals and humans. Through the second goal, “Greater efforts to prevent infections and to facilitate antibiotic alternatives”, the immediate environment of humans and animals such as surfaces is also mentioned. Here, it is emphasised to execute thorough hygiene measures to prevent the spread of AMR from the environment. The third goal, “Enhanced knowledge to improve targeted measures”, admits the need to promote knowledge building on the impacts of the environment [50]. In the report, the European Union action plan is highlighted, which integrates a One Health approach to tackle AMR. In the report, the role of the environment is emphasised as an area in need of engagement. The environmental role for transmission, potential tools and methods will be considered as well as data from environmental monitoring programmes [15,50].

### 3.2. Aflatoxin M1 in Italy as a One Health Issue

Several aflatoxin “crises” have occurred in northern Italy, the first and most severe in 2003 and the last happened from 2015 to 2017. These events were characterised by highly increased levels of Aflatoxin B1 (AFB1) in corn used for feed, and of Aflatoxin M1 (AFM1) in milk and dairy products. This happened in relation with environmental conditions featuring high temperatures, drought and enhanced insect damage of the crop [51,52,53].

Aflatoxins rank prominently among mycotoxins because of their genotoxic potential. AFB1 can cause hepatocellular carcinomas in humans, a type of liver cancer, with a higher risk for people infected with the hepatitis B or hepatitis C virus. EFSA considers that current levels of exposure to aflatoxins in foods may represent a health concern, in particular for younger age groups. In Europe, the food categories “liquid milk” and “fermented milk products” have been identified as the main contributors to overall AFM1 exposure throughout all age groups, infants being most exposed. Legal limits of foods and feeds and official monitoring programs are in place to prevent the risks for human and animal health due to aflatoxins. In the European Union, the legal limit for AFM1 in dairy products is 0.05 microgram per kilogram [54]. Aflatoxin production by fungi are common in hot and humid climate conditions, and can take place pre- and post-harvest [40]. In Serbia, a study showed that changes in temperature and moisture, resulting in the alternation of drought and flooding, enhance aflatoxin production. Hence, climate change may increase the health risks due to aflatoxin contamination of food [55].

AFB1 affects mainly grains and nuts, which are also the main sources of human exposure. However, the contamination of crops, such as corn used as animal feed, lead to the intake and digestion by farm animals. Dairy-producing ruminants transform AFB1 into AFM1, which is also a public health hazard, because it is genotoxic and carcinogen in vivo, even though it is less potent than AFB1 with respect to liver carcinogenicity. The toxic metabolite resulting from feed contamination is found in dairy products of ruminants like cattle, sheep, goats and buffaloes [40,51]. Dairy products are an important component of the diet in Italy [56]. AFM1 binds with proteins in milk and therefore, concentrates in cheese and other dairy products with a high protein content, such as the whey-based ricotta [57]. The National Reference Centre for the quality of bovine milk recommends that control of cheeses are postponed as compared to milk in consideration of the maturing periods of cheeses [53].

The area with the highest milk production is the Po Valley in northern Italy, and it is among the foremost agriculture intensive areas in Europe. The different environmental farming conditions of low- and high-yield dairy cows have an impact on AFM1 contamination. In low-yield cows, the carry-over of AFM1 to milk is in the 0.1–0.5% range of the AFB1 intake, but it is 1% to up to 6% in high-yield cows [57,58,59]. The environmental and agricultural scenario in Italy makes the area with the highest dairy production the most vulnerable to AFM1 contamination. The climatic conditions are characterised by high humidity rates, averaging at about 80%. Climate changes lead to greater stress on the crops due to temperature increase alternating between drought and heavy rainfall. This leaves the crops vulnerable to aflatoxin-producing fungi [55,60]. In Italy, almost 95% of total milk production, 13.3 million tonnes in 2019, is provided by cattle [61]. The milk production shows a seasonal trend, being higher from March to May.

The latest data provided by the National Reference Centre for the quality of bovine milk showed that the climatic trend in late spring and early summer is the critical factor influencing the extent of AFB1 contamination in cereal and corn crops [54]. This trend has been confirmed by the data analysis since 2012. Hence, climate trends influence the extent of the contamination in feed, flour and silage entering the animal feed circuit during the summer and for the following twelve months. Thanks to prevention and control measures, the latest data do not indicate health concerns, as the samples collected in 2019 show a sharp decrease of AFM1 concentrations compared to samples found in the period from 2012 to 2016. This is clearly reflected by the percentage of samples above the legal limit: while from 2012 to 2016, the average of samples above the legal limit was 2.50% with a peak of 5.06% in 2016, in 2019 the non-compliant samples have been a mere 0.34% [54].

In southern Italy, the climate affects aflatoxin occurrence as well, although concentrations in milk are generally low in this area due to lower humidity, less intensive farming and lower milk yield per animal. However, AFM1 contamination was significantly higher in cold season, particularly in autumn, than in the warmer season of spring. In this scenario, the non-compliance with the legal limit was just 0.1% [56].

The crisis of 2003 has prompted the Italian Ministry of Health to issue a contingency plan for the prevention and risk management of aflatoxins contamination in the dairy chain and in the production of corn for human and animal consumption in extreme climatic condition [51]. Besides this, the regular monitoring of raw milk and feed, more intensive in vulnerable months, allows timely advice given to the dairy farms to launch corrective measures [52,53].

## 4. Discussion

The two case studies illustrate how the environment interacts with the health of humans and animals, making up an essential pillar of One Health. Indeed, environment-related factors can play multiple roles that need proper characterisation to manage the complexity inherent to One Health issues. In the following, the two cases will be categorised under the established scenarios (Table 1) to highlight the integral part of the environment within the cases.

### 4.1. Climate Change Modulating Risk Factors for Health

The Italian aflatoxin case illustrates well how toxic pollutants fit into the One Health context, bringing together human health, animal health and their products as well as the environment [9]. Aflatoxins are carcinogens, thus human exposure has major health implications [40]. While the main aflatoxin, AFB1, is a contaminant of foods of vegetable origins, dairy-producing ruminants transform it into the toxic metabolite AFM1, which is excreted in milk, representing an additional route of human exposure [51]. The environment has a crucial role, shaping the exposure scenarios and the consequent human risk: Climate influences the contamination of crops used for feed by aflatoxigenic fungi, as the AFM1 presence in milk is closely related to yearly climate patterns as well as to seasonality [54,55,56,60]. Further, the farming environment is important, with intensively bred, high-yield herds showing a greater carry-over of AFM1, even at comparable feed contamination levels [56,57,58,59].

The AFM1 case study highlights some noticeable implications. A number of economic drivers orient a large part of the dairy production of the Po Valley toward high-quality products like the made-in-Italy cheeses Parmigiano and Grana. These meet high demand from national and international markets, but require high-yield cows and high costs to maintain the technologically developed intensive farming [51]. This economic trend makes the dairy farming system of the Po Valley more vulnerable to climate changes and associated risks such as AFM1 outbreaks.

This case study exemplifies the far-reaching impact of the environment in which feed is grown, from a One Health perspective. Although the current data indicate a low or very low carcinogenic risk from AFM1 in Italian dairy products, changes in the climate, as well as potential health hazards, justify continuous monitoring, crisis preparedness and regular updates of the exposure assessment [52]. In particular, modelling climate trends can aid to detect potential risks for aflatoxin occurrences, as a rise of AFM1 in milk is expected to occur from August to November due to the presence of AFB1 in feed materials in spring to early summer. Consequently, the sampling plan of feed and milk has to concentrate on this critical period [53]. In the face of a changing climatic scenario and potential following crises, the AFM1 issue has been efficiently managed through a food chain approach by the Italian Ministry of Health. This resulted in progressive reduction of the chance of consumer exposure [52,53].

### 4.2. The Anthropogenic Environment as a Source of One Health Risk Factors

Anthropogenic activities have led to new challenges for the environment [19]. Hence, complex issues like AMR must be handled in a coordinated manner. In the latest DANMAP report of 2019, the need to “supporting decision making in the prevention and control of resistant bacterial infections” was highlighted [44]. This requires an integrated approach tackling the complexity of AMR. Accordingly, the programme involves researchers from different disciplines, holding regular meetings between the Statens Serum Institut and the National Food Institute. Involved are veterinarians and public health professionals, such as physicians and epidemiologists but also microbiologists, which contributes to the farm-to-fork and One Health approach [48]. By including various disciplines, a wide range of expertise comes together, which can constantly improve the DANMAP. Additionally, researchers, political actors (Danish Ministry of Environment and Food and Ministry of Health) and private stakeholders from relevant sectors (e.g., pharmaceutical industries, meat chain enterprises, as well as farmers, retail, feed mills, pharmacies, etc.) are also continuously involved. For example, private stakeholders are engaged, as data are obtained from feed mills, slaughterhouses and via samples from food for human consumption [44]. This strengthens trust between the parties and has likely facilitated the large amount of voluntary data that is produced by the industry. To improve transparency, the DANMAP reports or website can provide additional information on the engagement of the public, consumers and the media.

The involvement of researchers from different disciplines and the cooperation among veterinary, food, human and environmental laboratories in terms of data sharing and common technological platforms are proficient ways to integrate the environment sector into AMR surveillance [44,48]. Additionally, the Danish One Health strategy to tackle antibiotic resistance and European approaches for AMR and One Health are good bases for establishing a connection to the environment and strengthening environment-related research for these topics [12,15]. For DANMAP, strengthening environmental research can facilitate the integration of environmental considerations into its analysis. These can encompass areas such as antibiotic use in plants, pesticides, manure and wastewater.

For instance, plant agriculture frequently uses antibiotics to enhance crop yields. This means that fruits and vegetables can be a source for AMR [62,63]. Pesticides may be a pathway for AMR, as some chemical substances may exert a selective pressure favouring antibiotic resistant bacteria [19]. In the Danish agricultural practice, the use of highly toxic and persistent substances is severely restricted. For instance, the insecticide lindane was been banned in Denmark since 1994. While the ban of high-concern pesticides is beneficial to humans, animals and ecosystems, these substances may leave environmental “legacies”. In the case of lindane, the by-product β-hexachlorocyclohexane can still be found in soils and wastewater, as it resists not only germs but also biodegradation, posing risks to human health [64]. Most important, there are indications that pesticides, their residues and by-products may increase the presence of AMR in the environment [65]. Some herbicides like glyphosate and glufosinate represent telling examples of widespread chemicals with the potential to increase the environmental AMR burden [38]. The overall use of substances in both animal and plant farming, including the overuse and misuse of antimicrobials as well as some pesticides can therefore act in an additive way [19,65]. Another environmental factor to be considered is the contribution to AMR by toxic metals, which can derive from soil composition, industrial emission or, in the case of copper, also from its use as feed additive [11,66].

However, more data are needed to conduct a meaningful risk assessment that comprehensively considers these environmental factors and weighs their possible contributions. Continuous monitoring and assessments must be maintained to prevent AMR and toxic by-products entering the ecosystem. Readopting sampling and screening measures within DANMAP for fruits and vegetables can aid in determining the current role of AMR and pesticides.

Further, monitoring manure used on soils is essential to screen AMR and infectious agents to prevent the spread into the food chain [34,67]. In connection with manure, wastewater is an important variable in the distribution of AMR and resistant pathogens. While the DANMAP reports acknowledge water as a source for resistant pathogens, more effort can be put into implementing water monitoring, as resistant pathogens can spread through use of wastewater, consumption of water and contamination of food or kitchen utensils [34,62]. AMR can persist for a long time in wastewater of plant and animal agriculture, and intensive animal farming may lead to a greater environmental enrichment of AMR [67,68]. The surveillance of soil and wastewater in water treatment plants, which turn wastewater into drinking water, is crucial to mitigate risks of infection or AMR [22]. For a comprehensive understanding of AMR, it is important to identify overuse misuse as well as critical pathways, and to recognise the connections between soil, manure and water to gauge the anthropogenic impact. One of DANMAP’s objectives is to explore further research areas and this could include investigations into plants, soil and water. These investigations can aid in determining any inadequate use of antimicrobials in agricultural settings and fuel the search of alternatives to bioaccumulating toxins and pesticides. This can support a surveillance approach that is holistic and foster research and development of environmental effects on AMR.

In the case of DANMAP, the farm-to-fork approach and the One Health strategy are integrated into the programme. The reoccurring emphasis of the One Health approach can strengthen the inclusion of all disciplines. Nevertheless, it is important to consider that this does not necessary entail that all disciplines must be represented equally in each scenario. Transmission routes for AMR-carrying microbes occur more often through contacts between animals, their products and humans rather than through the environment [69,70]. Hence, the veterinary and public health disciplines have a paramount role in this field. Nevertheless, environmental factors doubtlessly modulate AMR transmission as, for instance, AMR-carrying bacteria from animal farms persist for a long time in water, even after going through wastewater treatment plants [67]. The DANMAP reports consistently refer to the need of complying, upholding and improving current surveillance and prevention measures for infections through resistant bacteria [44]. Hence, it is crucial to foster the engagement of the environment to a necessary degree to characterise the environmental transmission of AMR in a qualitative as well as quantitative way, and to establish preventive measures.

### 4.3. Way Forward for the Environment and One Health

The two case studies show that the assessment of environmental risk factors is relevant to One Health surveillance. The accumulation of toxins from fungi, pesticides, manure or other sources in the environment can have downstream effects on human and animal health. Food as well as feed safety and surveillance are important to detect foodborne diseases and harmful accumulated chemicals. A structured analysis based on the identification of points of particular attention can support surveillance activities. Under this respect, Lombardo et al. have proposed a scheme for the analysis of environment-related factors in the animal farming scenarios with a One Health view. The proposed system considers the area (geo-climatic factors, waste disposal sites, land usage, main crops, water sources) and farm characteristics (size and conditions, biosecurity, use and disposal of biocide and drugs, feed quality and origin) [71]. In addition, available information such as routine controls and previous alerts should be exploited and integrated in the scheme.

The surveillance of terrestrial and aquatic ecosystems are of increasing interest and relevance in the One Health approach, as knowledge of the ecology of organisms can help to model and predict recurrent threats. Examples include, but are not limited to, blooms of toxic algae and outbreaks of infections spilling out from wildlife like bats, to humans [72,73]. In these examples, the environmental expertise can support the epidemiological modelling by identifying relevant modulating factors, such as pollution and land use for algal blooms, and bat-borne infection, respectively.

The environment can encompass water or soil, but it can also cover less obvious areas such as slaughterhouses or other areas where food is produced and processed, as it was exemplified by the DANMAP and the AFM1 cases. Through these different environments, humans and animals are in some ways always connected, which highlights the importance of finding ways to integrate environmental perspectives via engaging experts, employing techniques to assess environment-related factors and sharing data. The One Health approach provides an essential tool to link various disciplines, and to investigate the specifics and added values of each field. Not missing out on the environmental pillar will benefit the One Health approach through opportunities for environmental research that aid to better understand links between humans, animals and the ecosystem. Additionally, the Covid-19 pandemic points out that the health impact of an infection can be significantly modulated by a number of environmental factors [28,29,30,31]. The view of Covid-19 as “syndemic” recognises the need to interpret and assess the complex interplay between an infectious agent and concurrent determinants related to the physical and social environment, which is consistent with the One Health approach [74].

## 5. Conclusions

One Health is an approach to assess and manage complex public health issues that are cross-cutting and require the cross-fertilisation and integration of different expertise [1,2]. Therefore, One Health links the environment, humans, animals, including the food and feed chain. One Health approaches can be modulated in a case-specific way, as not all sectors need always be involved to the same degree.

International engagements like the Tripartite or European approaches must continue to refer to One Health, while also emphasising the importance of the environment pillar of the One Health approach.

In the AFM1 case study, environmental components are represented mainly by climate patterns and by the more or less intensive dairy farming scenarios in different Italian areas. These determinants are directly influencing the extent of contamination of feed by AFB1 and of milk by AFM1, and thus are directly linked to the AFM1-associated risk to human health [51,52,53,55,56].

In the AMR case study, the environment is not the main area of focus of the DANMAP, but nonetheless important, as anthropogenic activities contribute to the flow of bacteria-carrying AMR from sources like hospitals and farms. Potential overuse and misuse of AMR contribute to the occurrence of AMR in the environment, in particular soil and water, which are reservoirs for animals and humans [62,63,67].

One Health is a developing, multifaceted web of feedbacks and interactions among its components. The goal is not to drown in complexity, but to manage complexity. Further work is needed to better define the roles of environment in One Health scenarios. The characterisation of environmental factors is paramount to model the risks for animal and human health. One Health should be implemented as an institutional tool in public health, especially fit for evidence-based priority setting and to support decision-making [4,20]. More case studies are needed to showcase the role of the environment, highlighting the benefits of environmental expertise in connection to human and animal health.

## Figures and Tables

**Figure 1 medicina-57-00240-f001:**
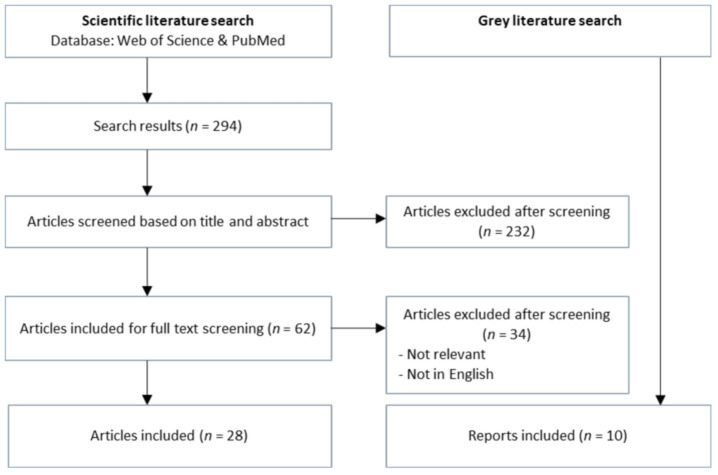
Screening process of literature from the databases Web of Science and PubMed.

**Table 1 medicina-57-00240-t001:** Examples of environmental scenarios of One Health relevance.

Scenario	Example of Risk Factors	Implications
**Environmental changes modulating risk factors for health** Climate change contributes to the distribution of insect vectors of zoonotic agents and to the increased amount of bioavailable toxicants in the environment. These toxicants accumulate in food-producing organisms (plants, animals).	Zoonotic vectors living in warmer areas; Occurrence of toxic metals; Meteorological conditions, air pollutants, sewage, wastewater and industrial chemicals.	Through insect migrations, arthropod-borne zoonoses can spread to colder world area [7]; Toxic metals like lead cause adverse health effects for animals and humans, in particular affecting the nervous system [27]; Infectious diseases (like Covid-19) modulated by environmental factors, including immunotoxic chemicals [28,29,30,31].
**Anthropogenic activities as a source of One Health risk factors through the environment** Farming activities may release noxious emissions, waste and by-products into the environment, which affect ecosystem quality, animal and human health. Industrial and other environment-modifying human activities affect food-producing organisms, thereby causing human exposure to hazardous agents.	Agricultural waste and by-products; Industrial emissions; Polluted wastewater or manure; Persistent, bioaccumulating substances (e.g., the by-product β-hexachloro-cyclohexane); Herbicides (e.g., glyphosate and glufosinate).	Agricultural waste and by-products can affect ecosystems, animal and human health, either directly or indirectly by contributing to climate changes [34]; Bioaccumulation of pollutants and by-products exposes farm animals and subsequently humans to toxic substances [35]; Increase of selective pressure towards antibiotic resistance of bacteria [38].

## Data Availability

Data sharing not applicable. No new data were created or analyzed in this study. Data sharing is not applicable to this article.

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
