# Peer review of "Assessing Environmental Factors within the One Health Approach"

_medicina, 2021, doi:10.3390/medicina57030240_

Round 1
Reviewer 1 Report
Reviewer’s Comments
Name of journal: Medicina (ISSN 1010-660X)
Manuscript ID: medicina-1107798
Type: Review
Title: Assessing environmental factors within the One Health approach
Summary:
This review highlights the importance of the environmental pillar in the One Health approach, through two case studies investigating the Danish antimicrobial resistance monitoring program and aflatoxin cases in Italy. Discussions highlight the importance of antimicrobial management in livestock and agriculture, which is key to preventing the increase in antimicrobial resistance, and ultimately, climate change affecting aflatoxin production, highlighting the increase in this risk factor in the dairy supply chain. This review highlights interesting environmental issues and the importance of a surveillance program, which takes a closer look at environmental risk factors, unfortunately increasing due to man. I find the proposed review interesting.
Specific comments:
1) Line 161-162: « PubMed using the search term “Aflatoxin M1” AND (“dairy” OR “cheese” OR “milk”) ». Please correct it to « search term “Aflatoxin M1” and (“dairy” or “cheese” or “milk”) ».
2) Line 169-170: « The search identified 294 articles, of which 28 articles were included in the analysis, see Figure 1. In total, 275 articles were excluded based on a screening ». Check the number of items excluded, I think they are 276 and not 275.
3) Line 188: « The DANMAP presents four objectives: The first two relate », please correct it to « The DANMAP presents four objectives: the first two relate ».
4) Line 236-237: « In Europe, the food categories ‘liquid milk’ 237 and ‘fermented milk products’ have been», please correct it to « In Europe, the food categories “liquid milk” 237 and “fermented milk products” have been ».
Author Response
Dear reviewer 1,
Thank you very much for your useful review. Please find our detailed answers to your comments in the table below. Please be aware that the lines indicated in the column Authors replies refer to the revised manuscript.
Best regards,
Sarah Humboldt-Dachroeden & Alberto Mantovani
|
Comments by reviewer 1
|
Authors replies |
|
This review highlights the importance of the environmental pillar in the One Health approach, through two case studies investigating the Danish antimicrobial resistance monitoring program and aflatoxin cases in Italy. Discussions highlight the importance of antimicrobial management in livestock and agriculture, which is key to preventing the increase in antimicrobial resistance, and ultimately, climate change affecting aflatoxin production, highlighting the increase in this risk factor in the dairy supply chain. This review highlights interesting environmental issues and the importance of a surveillance program, which takes a closer look at environmental risk factors, unfortunately increasing due to man. I find the proposed review interesting. |
Thank you very much! |
|
Line 161-162: «PubMed using the search term “Aflatoxin M1” AND (“dairy” OR “cheese” OR “milk”)». Please correct it to «search term “Aflatoxin M1” and (“dairy” or “cheese” or “milk”)». |
Thank you for this observation, we have corrected the sentence in line 171. |
|
Line 169-170: « The search identified 294 articles, of which 28 articles were included in the analysis, see Figure 1. In total, 275 articles were excluded based on a screening ». Check the number of items excluded, I think they are 276 and not 275. |
Thanks for catching this mistake. In fact, the number of excluded articles was wrong, 266 articles were excluded, which is corrected in line 179. |
|
Line 188: « The DANMAP presents four objectives: The first two relate », please correct it to « The DANMAP presents four objectives: the first two relate ». |
Thank you for this observation, we have corrected the sentence in line 197. |
|
Line 236-237: « In Europe, the food categories ‘liquid milk’ 237 and ‘fermented milk products’ have been», please correct it to « In Europe, the food categories “liquid milk” 237 and “fermented milk products” have been ». |
Please find the corrections in lines 245-246. |
Reviewer 2 Report
Review
The manuscript “Assessing environmental factors within the One Health approach” deals with a topic of global relevance and interest, which always deserves attention and awareness, and therefore such kind of scientific publications are welcome.
Some concerns of this paper stands on its overall structure: some chapters are quite redundant and repetitive; the concepts are absolutely important, but are melted into and with others. In the Introduction section, the topic could be discussed in a broader context and with specific examples ( possibly in Introduction and Discussion sections)
There are no specific data in the numbers that should be quoted on the basis of the references, there are only general statements, e.g. lines 270-274,
The title of the chapter 4.2. “The anthropogenic environment as a source of One Health risk factors” refers to anthropogenic factors, but only one of them - AMR, is mentioned in the section, and there is no
Important aspects such as the growing phenomenon of bacterial antibiotic resistance as a result of the abuse and inadequate use of antibiotics were not addressed in the Review.
Another general remark: I consider that the manuscript needs a revision of the English; examples: line 47- adaption (adaptation), line 195- not neglected ( neglected ) 228 – crises (crisises). Also, English grammar should be improved.
The authors of manuscript are asked to consider the above comments, make appropriate corrections, and I recommend the article for publication.
Author Response
Dear reviewer 2,
Thank you very much for your useful review. Please find our detailed answers to your comments in the table below. Please be aware that the lines indicated in the column Authors replies refer to the revised manuscript.
Best regards,
Sarah Humboldt-Dachroeden & Alberto Mantovani
|
Comments by reviewer 2 |
Authors replies
|
|
The manuscript “Assessing environmental factors within the One Health approach” deals with a topic of global relevance and interest, which always deserves attention and awareness, and therefore such kind of scientific publications are welcome. |
Thank you very much |
|
Some concerns of this paper stands on its overall structure: some chapters are quite redundant and repetitive; the concepts are absolutely important, but are melted into and with others. In the Introduction section, the topic could be discussed in a broader context and with specific examples (possibly in Introduction and Discussion sections) |
We thank the reviewer for this comment. The text was revised and more examples, in particular concerning climate changes, AMR, and One Health aspects of the Covid-19 pandemic were included. Additions were made to the introductory and discussion parts as recommended. A number of new and recent references have also been added to strengthen the paper’s content and message. Besides, a few mistakes (reference numbering) have been corrected in the aflatoxin case study. Please find the changes in the following lines: lines 47-48; lines 123-126; lines 142-144; lines 381-386; lines 412-417; and lines 426-430. |
|
There are no specific data in the numbers that should be quoted on the basis of the references, there are only general statements, e.g. lines 270-274, |
Thank you for this comment. In the section, we clarified some statements and added the numbers (%) to the new version. Please find the changes in lines 247-251; lines 281-284; and lines 288-289. |
|
The title of the chapter 4.2. “The anthropogenic environment as a source of One Health risk factors” refers to anthropogenic factors, but only one of them - AMR, is mentioned in the section, and there is no Important aspects such as the growing phenomenon of bacterial antibiotic resistance as a result of the abuse and inadequate use of antibiotics were not addressed in the Review. |
Thank you for detecting this important issue. In the chapter 4.2. “The anthropogenic environment as a source of One Health risk factors”, we have added some sentences elaborating anthropogenic activities and mentioned the topic misuse and overuse of antimicrobials. Please find these additions in lines 329-330; lines 361-369 and lines 381-386. Please also see the changes in the conclusion in lines 443-446. |
|
Another general remark: I consider that the manuscript needs a revision of the English; examples: line 47- adaption (adaptation), line 195- not neglected (neglected) 228 – crises (crisises). Also, English grammar should be improved. |
Line 49-50: adaption was changed to adaptation
Line 204: It was intended to write “not neglected”. However, to avoid the double negative, we re-wrote the sentence.
Line 237: We do want to refer to the plural form of the word crisis, hence, “crises” is the correct word. |
Reviewer 3 Report
The major fault with the manuscript is glossing over two major factors in evaluating the effect of environment on animal and human health: (1) the overuse of antibiotics in farm animals, and (2) the widespread use of large quantities of agrochemicals such as glyphosate and genetically modified crops, both of which are strongly predicted to affect microbes in the soil, water, plants, animals and humans. In-depth discussion of these environmental factors, their prevalence in animal feed and downstream human food, and effects on animal and humans would greatly contribute to the manuscript.
Author Response
Dear reviewer 3,
Thank you very much for your useful review. Please find our detailed answers to your comments in the table below. Please be aware that the lines indicated in the column Authors replies refer to the revised manuscript.
Best regards,
Sarah Humboldt-Dachroeden & Alberto Mantovani
|
Comments by reviewer 2 |
Authors replies
|
|
The major fault with the manuscript is glossing over two major factors in evaluating the effect of environment on animal and human health: (1) the overuse of antibiotics in farm animals, and (2) the widespread use of large quantities of agrochemicals such as glyphosate and genetically modified crops. Both of which are strongly predicted to affect microbes in the soil, water, plants, animals and humans. In-depth discussion of these environmental factors, their prevalence in animal feed and downstream human food, and effects on animal and humans would greatly contribute to the manuscript. |
Thank you for your comment. We fully concur with the reviewer that pesticides (incl. glyphosate), and feed additives (e.g. copper) should be considered in a comprehensive manner regarding AMR, also together with environmental pollution by heavy metals. We consider that an in-depth evaluation of the impact of environmental toxicants on AMR is beyond the scope of the paper, and it would require a stand-by-itself paper. However, we took the reviewer’s suggestion, by giving more emphasis to this topic and by adding relevant references.
For example, in lines 142-144, we used glyphosate as an example in the introduction. In the discussion, we also added some elaborations. In chapter 4.2., we added paragraphs and references in terms of glyphosate as well as overuse of antibiotics in agriculture, see lines 362-369 and lines 381-386. In chapter 4.3., lines 402-406, discussions about downstream effects of toxins have been added. |
Round 2
Reviewer 3 Report
The authors have addressed my concerns.